# Oral Dysbiosis Is Associated with the Pathogenesis of Aortic Valve Diseases

**DOI:** 10.3390/microorganisms13071677

**Published:** 2025-07-16

**Authors:** Erika Yaguchi, Yuske Komiyama, Shu Inami, Ikuko Shibasaki, Tomoaki Shintani, Ryo Shiraishi, Toshiki Hyodo, Hideki Shiba, Shinsuke Hamaguchi, Hirotsugu Fukuda, Shigeru Toyoda, Chonji Fukumoto, Sayaka Izumi, Takahiro Wakui, Hitoshi Kawamata

**Affiliations:** 1Department of Oral and Maxillofacial Surgery, Dokkyo Medical University School of Medicine, 880 Kitakobayashi, Shimotsuga, Mibu 321-0293, Tochigi, Japan; eri-yagu@dokkyomed.ac.jp (E.Y.); y-komi@dokkyomed.ac.jp (Y.K.); ryo-s@dokkyomed.ac.jp (R.S.); hyodo14@dokkyomed.ac.jp (T.H.); chonji-f@dokkyomed.ac.jp (C.F.); saya@dokkyomed.ac.jp (S.I.); 2Department of Cardiovascular Medicine, Dokkyo Medical University School of Medicine, 880 Kitakoba Yashi, Shimotsuga, Mibu 321-0293, Tochigi, Japan; s-inami@dokkyomed.ac.jp (S.I.); s-toyoda@dokkyomed.ac.jp (S.T.); 3Department of Cardiac and Vascular Surgery, Dokkyo Medical University School of Medicine, 880 Kitakobayashi, Shimotsuga, Mibu 321-0293, Tochigi, Japan; sibasaki@dokkyomed.ac.jp (I.S.); fukuda-h@dokkyomed.ac.jp (H.F.); 4Center for Oral Clinical Examination, Hiroshima University Hospital; 1-2-3 Kasumi, Minami-ku, Hiroshima 734-8551, Hiroshima, Japan; tshintan@hiroshima-u.ac.jp; 5Department of Biological Endodontics, Graduate School of Biomedical and Health Science, Hiroshima University, 1-2-3 Kasumi, Minami-ku, Hiroshima 734-8553, Hiroshima, Japan; bashihi@hiroshima-u.ac.jp; 6Department of Anesthesiology, Dokkyo Medical University School of Medicine, 880 Kitakobayashi, Shimotsuga, Mibu 321-0293, Tochigi, Japan; s-hama@dokkyomed.ac.jp

**Keywords:** oral dysbiosis, aortic valve disease, infectious endocarditis, 16S rRNA metagenome analysis

## Abstract

The involvement of oral bacteria in the pathogenesis of distant organs, such as the heart, lungs, brain, liver, and intestine, has been shown. We analyzed the distribution of bacterial species in the resected aortic valve by 16S rRNA metagenomic analysis and directly compared their gene sequences with those in the oral cavity. Thirty-two patients with aortic stenosis or aortic regurgitation who underwent aortic valve replacement were enrolled in this study. Antibody titer against periodontal pathogenic bacteria in the patient’s serum was analyzed. The genetic background and distribution of bacterial species on subgingival plaque, the dorsal surface of the tongue, and the resected aortic valve were analyzed. Patients with aortic valve disease were shown to have more severe periodontal disease by the detection of antibodies against Socransky’s red-complex bacteria of periodontitis. Bacterial DNA was detected in the aortic valves of 12 out of 32 patients. The genomic sequences of the V3-V4 region of the 16S rRNA in some bacteria isolated from the aortic valves of six patients who underwent metagenomic analysis were identical to those found in the oral cavity. The findings indicate that bacteria detected in the aortic valve may be introduced through oral dysbiosis, a condition characterized by an imbalance in the oral microbiota that increases the risk of periodontal disease and dental caries. Oral dysbiosis and the resulting potential bacteremia are associated with the pathogenesis of aortic valve diseases.

## 1. Introduction

The involvement of oral bacteria in the pathogenesis of distant organs, such as the heart, lungs, brain, liver, and intestine, has been shown [1,2]. Dysbiosis is known as pathological changes (type and/or proportion of bacteria) of commensal bacteria in some organs, which cause various diseases [3]. Oral dysbiosis is a condition characterized by an imbalance in the oral microbiota that increases the risk of periodontal disease and dental caries. Recently, several reports have demonstrated an association between oral bacteria or oral dysbiosis and diseases of other organs through metagenome analysis and molecular biological analysis [4,5].

One known disease closely associated with oral bacteria is infectious endocarditis (IE), which causes verrucae on the endocardium. Aortic valve disease (AV) is known to increase the risk of IE. Among aortic valve diseases (aortic stenosis (AS) and aortic regurgitation (AR)), AS is a condition in which blood flow from the left ventricle to the ascending aorta during systole is obstructed due to narrowing of the aortic valve orifice [6]. The causes include idiopathic degenerative sclerosis with calcification (congenital bicuspid valves prone to sclerosis) and rheumatic fever. Valve replacement or valvuloplasty is indicated when stenosis and regurgitation after AS/AR are severe [7]. As per the general concept of the pathogenesis of IE, the blood flow jets caused by the valvular diseases (AS and AR) damage the valves and endocardium, resulting in the formation of a sterile vegetation at the site of the damage. Bacterial infection on these vegetations is thought to lead to the development of IE.

We have previously reported that more than 60% of bacteria detected in the blood cultures of patients with IE are recognized as oral bacteria [8]. In our previous study, bacterial cultures of blood from patients with IE were compared with dental plaque cultures in the oral cavity to verify the morphological and biochemical identity. DNA sequencing of bacteria in blood and dental plaque from patients with IE can reveal their genetic identity. However, in only one case the bacteria detected in dental plaque and blood had a 100% identical 16S rRNA gene sequence [8]. Although this study showed indirect evidence for so-called focal dental infections in distant organs, it was difficult to genetically identify the causative bacteria by comparing the limited number of clones detected in the bacterial cultures.

In this study, we detected bacteria in the resected aortic valve and analyzed the distribution of bacterial species in the resected aortic valve by metagenomic analysis. Simultaneously, we analyzed the distribution of bacterial species in subgingival dental plaque, which causes periodontal disease, and in swabs from the dorsal surface of the tongue, which reflects the bacterial microbiota of the oral cavity in the same patients. Then we verified whether oral bacteria reached the aortic valve by directly comparing their gene sequences.

## 2. Materials and Methods

### 2.1. Patients

Thirty-two patients with AS or AR who underwent aortic valve replacement between 1 May 2020, and 31 March 2021, at Dokkyo Medical University Hospital were enrolled in this study. The inclusion criteria for this study were that the aortic valve was resected during the designated period and the material was aseptically available. No exclusion criteria were defined. The 32 patients included 14 patients with diabetes mellitus who were adequately controlled. Four patients out of thirty-two were taking steroids to control their rheumatoid arthritis or interstitial pneumonia. None of the patients had received antimicrobial therapy in a week prior to oral bacteria collection or in a week prior to aortic valve resection. Severity of the periodontitis was classified according to the classification of periodontitis by the Japanese Society of Periodontology [9]. This study was approved by the Ethics Committee of Dokkyo Medical University (approval no. R-16-14J and R-37-20J). Patients were included in this study only when consent was obtained from the patient or the key person. In cases where the patients themselves could not make a decision, we explained the details of this research to the key person in this study. In this study, we obtained genetic information from microorganisms, but not human-derived information.

### 2.2. Measurement of Serum Antibody Titer Against Periodontal Pathogenic Bacteria

Serum antibody titers were measured according to a previous report [10,11]. Serum from the preoperative blood collection of patients was isolated and stored at −80 °C. Serum was collected from patients during surgery and stored at −80 °C. Serum IgG antibody titers against periodontal pathogenic bacteria were determined by the enzyme-linked immunosorbent assay (ELISA) using strains of bacteria that had already been isolated and identified. Sonicated preparations of bacteria (*Porphyromonas gingivalis (Pg)*, *Tannerella forsythia (Tf)*, *Treponema denticola (Td)*, *Aggrigatibactor actinomycetemcomitans (Aa), Eikenella corrodens (Ec)*, *Fusobacterium nucleatum (Fn)*, *Prevotella intermedia (Pi)*, *Prevotella nigrescens (Pn)*, *Campylobactor rectus (Cr)*) were used as bacterial antigens. The sera from five healthy subjects (ages 12–81) were pooled and used for calibration. With serial dilutions of this pooled control serum, the standard reaction was defined as an ELISA unit (EU), so that 100 EU corresponded to 1: 3200 dilutions of the calibrator sample. The following formula was applied to the EU to calculate the diagnostic standardized value: (IgG titer of patient- mean IgG titer of healthy subjects)/2 standard deviations (SD) determined by mean IgG titer of five healthy subjects [10].

### 2.3. DNA Extraction from the Aortic Valve or Oral Bacteria

Aortic valves were stored in 1.5 mL microtubes at −80 °C immediately after surgical excision. Isospin Tissue DNA (Nippon Gene, Co. Ltd., Tokyo, Japan) was used to extract DNA from excised aortic valves. Samples of the dorsal surface of the tongue, representing the bacterial microbiota of the oral cavity, were collected using cotton swabs. The swabs were rubbed on the dorsal surface of the tongue and stored in conical tubes at −80 °C until DNA extraction. In addition, subgingival plaque samples were collected from sites with obvious inflammatory conditions, such as periodontitis, using a manual scaler and stored in 1.5 mL microtubes at −80 °C until DNA extraction. DNA was extracted from plaque and tongue swab samples using the Isoil DNA extraction kit following the manufacturer’s instructions (Nippon Gene, Co. Ltd., Tokyo, Japan). The plaques were directly added to the extraction buffer, and the swab tips were cut into the extraction buffer, thoroughly agitated, and centrifuged to obtain the supernatant.

### 2.4. Confirmation of Bacterial DNA in the Extracted Samples

Extracted DNA was confirmed using primers against bacterial 16S rRNA. The primers used were 16S rRNA-27f: 5′-AGAGAGTTTGATCCTGGCTCAG-3′ and 16S rRNA-1492r: 5′-ACGGCTACCTTGTTACGACTT-3′. We performed PCR on extracted DNA samples using QUICK Taq HS DyeMix (TOYOBO Co., Ltd., Osaka, Japan). The PCR products were subjected to 1.2% agarose gel electrophoresis to detect a band of approximately 1500 bp corresponding to 16S rRNA.

### 2.5. 16S rRNA Gene Amplicon Sequencing Analysis

For amplicon sequencing, libraries were prepared according to the Illumina 16S metagenomic sequencing library preparation protocols. A total of 2.5 µL of microbial genomic DNA (5 ng/µL in 10 mM Microbial genomic DNA 2.5 µL (5 ng/µL in 10 mM Tris pH 8.5) was used for the first PCR with the following primers: forward, 5′-TCGTCGGCAGCGTCAGATGTGTATAAGAGAGACAGCCTAHGGGRBGCAGCAG-3′, and reverse, 5′-GTCGTGGGCTCTCGAGAGATGTGTAGTAAGAGACAGGACTACHVGGGGTATCTAATCC-3′. PCR was performed using KAPA HiFi HotStart ReadyMix (KAPA Biosystems, Inc., Wilmington, MA, USA), and PCR products were analyzed using a Bioanalyzer DNA 1000 chip (Agilent Technologies, Inc., Santa Clara, CA, USA). PCR products were purified using an Agencourt AMPure XP 60 mL kit (Beckman Coulter, Inc., Brea, CA, USA). Then, a second PCR was performed to construct a sequencing library, following the Illumina 16S Metagenomic Sequencing Library Preparation Protocol. The PCR products were purified and subjected to amplicon sequencing. Sample sequencing was performed using Illumina MiSeq (Illumina, Inc., San Diego, CA, USA), and paired-end sequence data detected in each sample were stored in separate FASTA files for Read1 and Read2. Sequence data were analyzed using Qiime2 [12]. First, demultiplexing and denoising were performed to remove adapter sequences, linker sequences, and primer sequences.

### 2.6. Microbial Population Analysis

Before analysis by Qiime2, ASVs (amplicon sequence variants) were generated by filtering, denoising, chimera checking, and merging paired reads of the sequence data according to the DADA2 workflow [13]. The Silva dataset was used for the 16S rRNA taxonomy database [14,15], and the V3-V4 region of the 16S rRNA was cut from the full-length sequence to adjust the dataset for analysis. The alpha diversity of each bacterial microbiota and the beta diversity of the detected sites were examined.

### 2.7. Statistical Analysis

To examine the frequency of periodontitis in patients with aortic valve disease, a one-sample Z-test was used as a statistical test to compare the proportion of stage III + IV + edentulous patients with aortic valve disease with the proportion of stage III + IV + edentulous patients in the general population (63.5%), adjusted for age based on the results of the 2016 Survey of Dental Diseases of Japan. Standard errors were calculated based on the population proportions, and one-tailed tests were performed. The significance level was set at 0.05. Statistical analyses were conducted using R version 4.2.3 (R Foundation for Statistical Computing, Vienna, Austria) [16]. We examined the stages of periodontitis and antibody titers against periodontal pathogenic bacteria using the Kruskal–Wallis test. The correlation between the antibody titer and the stage of periodontitis was analyzed.

The alpha diversity of each bacterial microbiota and the beta diversity of the detected sites were examined. To assess the alpha diversity, Pielou’s evenness index, reflecting the evenness of taxonomic distribution rather than phylogenetic breadth, and Shannon’s diversity index, which accounts for both species richness and evenness, were calculated. A Kruskal–Wallis test was used to compare the distributions among plaque, tongue, and valve groups [17]. A significance level of α = 0.05 was used. The analysis was conducted using the diversity alpha plugin in Qiime2 (version 2021.11) and compared between groups by the diversity alpha-group-significance plugin.

To analyze the beta diversity, principal coordinate analysis (PCoA) was performed based on the distance and read count values in the phylogenetic tree of the detected bacteria [18,19]. We used PERMANOVA (permutational multivariate analysis of variance) to test for significant differences in microbial community composition among groups, based on the unweighted UniFrac distance matrix [20]. The analysis was conducted using the ‘adonis2′ function from the ‘vegan’ R package (version 2.6.4) [21], with 999 permutations.

## 3. Results

### 3.1. Periodontitis and Aortic Valve Condition in Patients with Aortic Valve Disease

Thirty-two patients with AS or AR who underwent aortic valve replacement between May 2020 and March 2021 at Dokkyo Medical University Hospital were enrolled in this study. The patients’ demographic data are presented in Table 1. These patients underwent an oral examination and oral management prior to surgery, and their periodontal disease was staged based on an intraoral examination, panoramic radiograph, and periodontal pocket depth [9].

Of the 32 patients, 15 were males and 17 were females, with an average age of 74.15 ± 8.172 years (Table 1). Five patients had periodontal disease stage I, two patients had stage II, five had stage III, ten patients had stage IV, and ten patients were edentulous. Eleven patients had more than 21 teeth, six patients had 11 to 20 teeth, and five patients had 1 to 10 teeth. Twenty-six patients had calcification in the aortic valves, and seven were suspected to have bicuspid valves based on transthoracic echocardiography. Twenty-five patients had severe AS, two had moderate to severe AS, three had moderate AS, one had severe AR, and one had moderate to severe AR. Bacterial DNA was detected in 12 patients (37.5%) from resected aortic valves by PCR in a total of 32 patients examined. Sufficient bacterial DNA was collected in six patients (50%) from the aortic valves for metagenomic analysis in the twelve patients examined. ASVs in the V3–V4 region of the 16S rRNA in all six patients (100%) matched between oral bacteria and bacteria collected from the aortic valves. Detailed data for individual patients are presented in Appendix A. AVAi averaged 0.45 ± 0.0012, SV averaged 78.93 ± 78.60, and SVi averaged 49.23 ± 27.57 (Appendix A).

We compared the severity of periodontitis in the patients in this study and of the same age group from the results of the Survey of Actual Conditions of Dental Diseases in Japan [22]; 36.5% of the patients were classified as having mild disease in stages I and II, while 25.9% of patients with aortic disease were in the same category. In contrast, the percentage of patients with advanced periodontitis in stages III and IV and the edentulous jaw group was 74.1% in patients with aortic valve disease and 63.5% in the Survey of Dental Diseases in Japan (Figure 1). Statistical comparisons were difficult because there was a large difference in population size (32 patients in this study and 2378 patients in the survey), and 32 patients were not extracted from 2378 patients. Although this is an exploratory analysis rather than a causal analysis, there was a tendency for more patients with aortic valve disease to have stage III or IV advanced periodontitis and edentulous jaws (*p* = 0.042, Z-test).

### 3.2. Measurement of Serum Antibody Titer Against Periodontal Pathogenic Bacteria in Patients with Aortic Valve Disease

Periodontal pathogenic bacteria are classified into several categories (color-coded as red, orange, yellow, green, blue, and purple complexes) based on their importance in clinical pathogenesis by Socransky et al. [23], and bacteria classified in the red complex (*Porphyromonas gingivalis*, *Tannerella forsythia*, *Treponema denticola*) are considered to be important for the pathogenesis of periodontitis (Figure 2A). The radar chart in Figure 2B shows the mean antibody titers of various periodontal pathogenic bacteria in the serum of patients with aortic valve disease. Antibody titers against bacteria in the red complex category were markedly elevated in patients compared to healthy controls. A comparison of serum antibody titers against red complex bacteria by periodontitis stage in patients with aortic valve disease showed that antibody titers tended to increase with the progression of the periodontitis stage, and antibody titers to red-complex bacteria were positively correlated with the periodontitis stage (Figure 2C). A weak but positive correlation was also observed between the antibody titer against the bacteria of the orange complex and the periodontitis stage (Figure 2C). Antibody titers against the bacteria of the blue and green complexes were weakly correlated with the periodontitis stage, although there was a tendency for antibody titers to increase.

### 3.3. Taxonomy Analysis

Bacterial 16S rRNA was detected in 12 aortic valves of 32 patients examined using PCR. We obtained sufficient DNA for amplicon sequencing analysis for six patients. Taxonomic analysis was performed to identify ASVs with >70% homology. The bacterial microbiota composition based on the read count of each ASV is shown in Appendix A. Surprisingly, several bacterial species were detected in the resected aortic valve. As expected, the bacterial microbiota of the tongue and periodontal pocket dental plaque were similar in many cases; however, the bacterial microbiota detected in the aortic valve showed a different pattern. *Proteobacteria*, *Fusobacteriota*, *Firmicutes*, *Bacteroidota*, and *Actinobacteriota* were found as the bacterial phyla with high read counts from the aortic valve (Appendix A). Some of the phyla were detected at similar frequencies at different sites in one patient, while others were detected at different frequencies in the aortic valve. In the present study, we did not search for individuals without periodontal disease or without aortic valve disease, so we were unable to verify whether or not oral dysbiosis occurs in patients with aortic valve disease by means of bacterial phyla detected on the tongue or in periodontal pockets. Furthermore, the organs in which the bacterial phyla detected in the aortic valve are endemic are diverse, and although many bacteria are endemic in the oral cavity, we cannot conclude that the oral cavity is a gateway for bacterial entry based on this alone. The same results were obtained, excluding the cases in which the second round PCR was carried out when performing metagenomic analysis. In general, the experiment using PCR carried the risk of sample cross-contamination, but the results of the analysis indicated that no cross-contamination of samples occurred.

### 3.4. α-Diversity and β-Diversity

Examination of α-diversity and β-diversity indicated that the bacterial microbiota in the resected aortic valve tended to be similar, although not identical to the composition of the oral bacterial microbiota. We examined the α-diversity of the plaques, tongue swabs, and excised aortic valves (Appendix A). There was no significant difference in the diversity within each bacterial microbiota detected at each site (*p* = 0.547). This suggests that a wide variety of bacteria are present in the resected aortic valve, and that, like the oral environment, the bacterial microbiota may be diverse. A Kruskal–Wallis test against Pielou’s evenness index revealed no significant difference among the three groups, H = 1.205, *p* = 0.547. Post hoc analysis using Bonferroni correction also showed no difference between groups (Appendix A, Appendix A). A Kruskal–Wallis test against Shanon’s index revealed no significant difference among the three groups, H = 0.889, *p* = 0.641. Post hoc analysis using Bonferroni correction also showed no difference between groups (Appendix A, Appendix A). When we examined the β-diversity among the bacterial microbiota (Appendix A), we found that the vectors of diversity of the bacterial microbiota differed among the three groups (*p* = 0.001): plaque, tongue, and resected aortic valve. Among the resected aortic valves, we recognized two groups with different vectors of diversity, and among the two groups, three cases, AS 14, 20, and 25, were considered to have bacterial microbiota similar to those of the oral-derived specimens, being closer to the oral bacteria. The frequency of detection of ASVs in AS 18, 28, and 30 was higher than that in the Unassigned group, which may have been a factor in the division into two groups. Although the Gene Analyzer detected the expected size of PCR products in these three samples at the first PCR stage, it was insufficient in quantity. As the primary purpose of this study was to examine whether bacteria were present in the resected aortic valve, we performed the first PCR again using the first PCR product as a PCR template. This increased nonspecific amplification, and unassigned ASVs became more frequent. However, from the viewpoint of Axes 2 and 3, there was a similar trend to that of the oral-derived specimens. A significant difference in microbial composition was observed among the three groups (PERMANOVA: *F* = 3.025, *R*^2^ = 0.287, *p* = 0.002) (Appendix A).

### 3.5. Characteristics of Bacteria Detected in the Tongue, Dental Plaque, and Resected Aortic Valve

The 195 bacteria detected in the tongue, dental plaque, and resected aortic valve specimens were ranked according to read count (Appendix A). The lists of the top 25 bacteria are shown in Appendix A. Among the bacteria detected in the resected aortic valve, oral commensal bacteria and periodontal pathogenic bacteria (color-coded as red, orange, yellow, green, blue, and purple) according to Socransky’s classification [23] were extracted and are shown in Figure 3. Among the bacteria listed for resected aortic valves, 48% (16–64%) of the 35 species were associated with the oral cavity, and 18% (4–28%) were periodontal pathogenic bacteria (Appendix A). In cases 20 and 25, red and orange bacteria were detected in the aortic valve. In case 30, neither red nor orange complex bacteria were detected in the aortic valve (Figure 3). In cases 14, 18, and 28, red and orange bacteria were detected in the aortic valve (Figure 3), although the bacterial microbiota detected on the dorsal surface of the tongue contained no red and only a few orange bacteria (Appendix A). In cases 20, 25, and 30, a significant number of red and orange bacteria were included in the bacterial microbiota considered endemic to the oral cavity detected on the dorsal surface of the tongue (Appendix A).

### 3.6. Identification of the Same ASV in the V3 -V4 Region of the 16S rRNA Sequence in the Tongue, Dental Plaque, and Resected Aortic Valve

The same ASV in the V3 -V4 region of the 16S rRNA sequence was detected in the tongue, dental plaque, and the resected aortic valve (Figure 4). This indicates that bacteria with the same genetic background, that is, strain-level similarity of the bacteria, are shown in the oral cavity and aortic valve. A total of 2524 independent ASVs were detected in all samples. Forty bacteria were found to have identical ASVs in the excised aortic valve and oral cavity (dorsal surface of the tongue and/or dental plaque). Fourteen bacteria were detected with identical ASVs from the three sites of the tongue, dental plaque, and excised aortic valve, and twenty-six bacteria were detected with identical ASVs from the excised aortic valve and tongue or dental plaque (Figure 4). Twelve of the forty bacteria were periodontal pathogenic bacteria, according to Socransky’s classification. Although the possibility that the bacteria detected from the aortic valve were transient adherences of oral bacteria due to bacteremia cannot be completely ruled out, it is considered extremely unlikely because of the intravenous administration of antibiotics during the perioperative period, the physical oscillation during the resection, and the saline rinse of the resected valve.

## 4. Discussion

In this study, we found several novel points: (1) patients with aortic valve disease were more likely to have periodontitis and more of them had severe disease or edentulous jaws; (2) serum titers of antibodies against periodontal pathogenic bacteria and red complexes were high in patients with aortic valve disease; (3) bacterial DNA was detectable in the resected aortic valve of 12/32 (37.5%) patients with aortic valve disease; (4) a wide variety of bacteria were detected in the resected aortic valves of six patients who underwent amplicon sequencing analysis; (5) 40 bacteria with strain-level similarity were found in both the resected aortic valve and oral bacterial ASVs; (6) among the 40 bacteria with strain-level similarity, 12 were periodontal pathogenic bacteria according to Socransky’s classification. These results strongly suggest that in patients with current periodontitis or previous periodontitis, oral bacteria continuously invade the bloodstream and settle in the aortic valve.

A potential limitation of this study is whether the fact that bacterial DNA was detected in the aortic valve in only 12 of 32 cases, and that sufficient bacterial DNA was recovered for metagenomic analysis in only 6 cases, is a result of manipulation or a genuine finding. In the future, it will be necessary to quantify the percentage of patients who have the same clone of oral bacteria in the aortic valve and to strictly control the detection limit of bacterial DNA from the valve to prevent cross-contamination. As this study is a PCR-based experiment, increasing the sensitivity reduces the specificity. Increasing the number of patients and setting a strict detection sensitivity threshold makes it possible to determine whether the detection frequency obtained in this experiment reflects the true situation or is merely a reflection of the detection sensitivity. Furthermore, since we are unable to examine the presence of oral bacteria in the aortic valves of healthy subjects or in patients with mild aortic valve disease, it is not known whether this observation is a phenomenon specific to patients with severe aortic valve disease. A study investigating the presence of bacteria in heart samples taken during autopsies of individuals without aortic valve disease who died in accidents could confirm whether the findings of this study are specific to patients with aortic valve disease. In AS 18, 28, and 30, samples at the first PCR stage were insufficient in quantity to construct a metagenome library. As the primary purpose of this study was to examine whether oral bacteria were present in the resected aortic valve, we performed PCR again using the first PCR product as a template. This increased nonspecific amplification, and unassigned ASVs became more frequent. However, examining the ASVs identified in these three patients reveals no evidence of contamination among the samples. In these three patients, the bacteria detected in the aortic valve included the same ASV as the oral bacteria, proving that they were bacterial with strain-level similarity in each patient.

In some cases (cases 20, 25, and 30), bacteria in the red and orange complexes were increased in the bacterial microbiota of the oral cavity. It was endemic to the oral cavity and was detected on the dorsal surface of the tongue, suggesting that dysbiosis had occurred. In cases 20 and 25, bacteria in the red and orange complexes were also detected in the aortic valve. In contrast, the dorsal surface of the tongue contained fewer orange complex bacteria in cases 14, 18, and 28, without red complexes. Nevertheless, red and orange complexes were detected in the aortic valve, suggesting that colonization of the aortic valve via bacteremia from periodontal bacteria in the periodontal pocket occurs even in the absence of oral bacterial dysbiosis. The similarity of the bacteria detected in the oral cavity (dorsal surface of the tongue or periodontal pockets) and the aortic valve was examined. In cases 14, 18, 20, and 25, a significant number of bacteria were detected from both sources. Only Staphylococcus and Neisseria were detected in cases 28 and 30, respectively, suggesting that oral bacteria may play a lesser role.

In many cases, Streptococcus, which appears to be an early colonizer and a subsequent dental plaque constituent bacteria, was also present in the aortic valve, suggesting that they may be involved in bacterial colonization. However, some ASV species that were detected in the resected aortic valve were not detected in the oral cavity. Because oral dysbiosis has been present for a considerable period, and the timing of bacterial invasion and settlement in the aortic valve does not always coincide due to repeated acute conversion and remission of periodontal disease, it is thought that there were bacteria that existed in the oral cavity but whose ASVs did not align.

Although numerous reports have documented the detection of oral bacteria in distant organs using PCR [8,24,25,26,27,28,29,30,31,32,33], few studies have confirmed whether the bacteria detected in the oral cavity and distant organs are the same clone [8]. For example, Zeibolz et al. examined the presence of pathogenic periodontal bacteria in aortic valves using PCR [24]. The relationship between oral bacteria and diseases of distant organs is also well known. Periodontal pathogens are known to be risk factors for atherosclerosis [25,26,27,28,29] and, as a result, are thought to be involved in the development of several cardiac diseases, including infective endocarditis [30]. Streptococcus mutans, a well-known bacterium that causes caries, is also involved in the development of cerebral hemorrhage [31,32,33]. Recent studies have demonstrated that oral bacteria can enter the bloodstream and colonize heart valves and blood vessels [34,35,36,37,38]. This may lead to chronic low-grade infections and promote inflammation and calcification in the vascular walls, potentially contributing to the development and progression of cardiovascular diseases. Although these findings are valuable as circumstantial evidence that oral bacteria can infect distant organs via the bloodstream, it is challenging to determine whether diseases in the oral cavity, such as periodontitis and dental caries, can serve as a gateway for entry. In this study, we found that bacteria with strain-level similarity, possibly the same clones, were present in the oral cavity, distant organs, and the aortic valve. However, at present, we have not clarified whether oral bacteria caused aortic valve disease or colonized the injured aortic valve.

The importance of oral management during surgery and several medical treatments has recently been reported [39]. However, there is no clear direct evidence that bacteria in the oral cavity can induce secondary infections in other organs. Accumulated experience strongly suggests that oral lesions (dental caries reaching the root canal or periodontal disease) may serve as a gateway for bacterial invasion into the body, and studies providing direct evidence have increased. In this study, we detected bacteria with strain-level similarity, that is, possibly the same clone of bacteria, in the oral cavities and resected aortic valves of patients with aortic valve disease. Oral dysbiosis and the resulting potential bacteremia are associated with the pathogenesis of aortic valve diseases.

## 5. Conclusions

Oral dysbiosis and the resulting potential bacteremia are associated with the pathogenesis of aortic valve diseases. Appropriate oral management may contribute to the prevention of aortic valve diseases. Further experiments may be necessary to demonstrate the direct association between oral bacteria and the onset or progression of aortic valve disease.

## Figures and Tables

**Figure 1 microorganisms-13-01677-f001:**
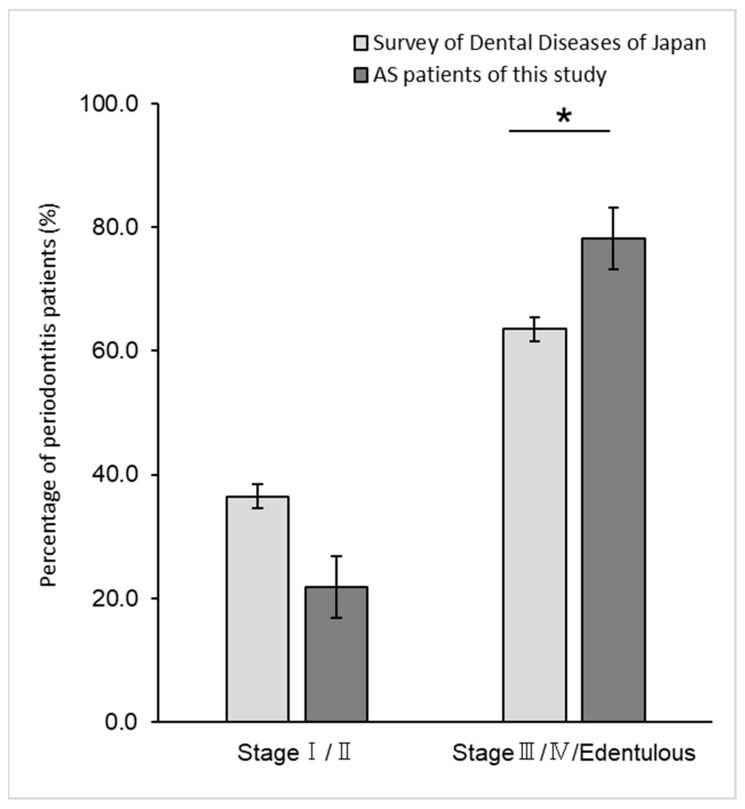
Periodontitis/edentulous status of patients with aortic valve disease. Stage represents the severity of periodontitis. Stages I and II were considered as early/mild periodontitis, while stages III and IV were considered as advanced and moderate to severe periodontitis. The edentulous patient was considered a result of periodontitis progression. There was a tendency for more patients with aortic valve disease to have stages III or IV advanced periodontitis and edentulous jaws (Z = 1.72, *p* = 0.042, one-tailed test). Bars indicate 95% confidence intervals. *: *p* = 0.042.

**Figure 2 microorganisms-13-01677-f002:**
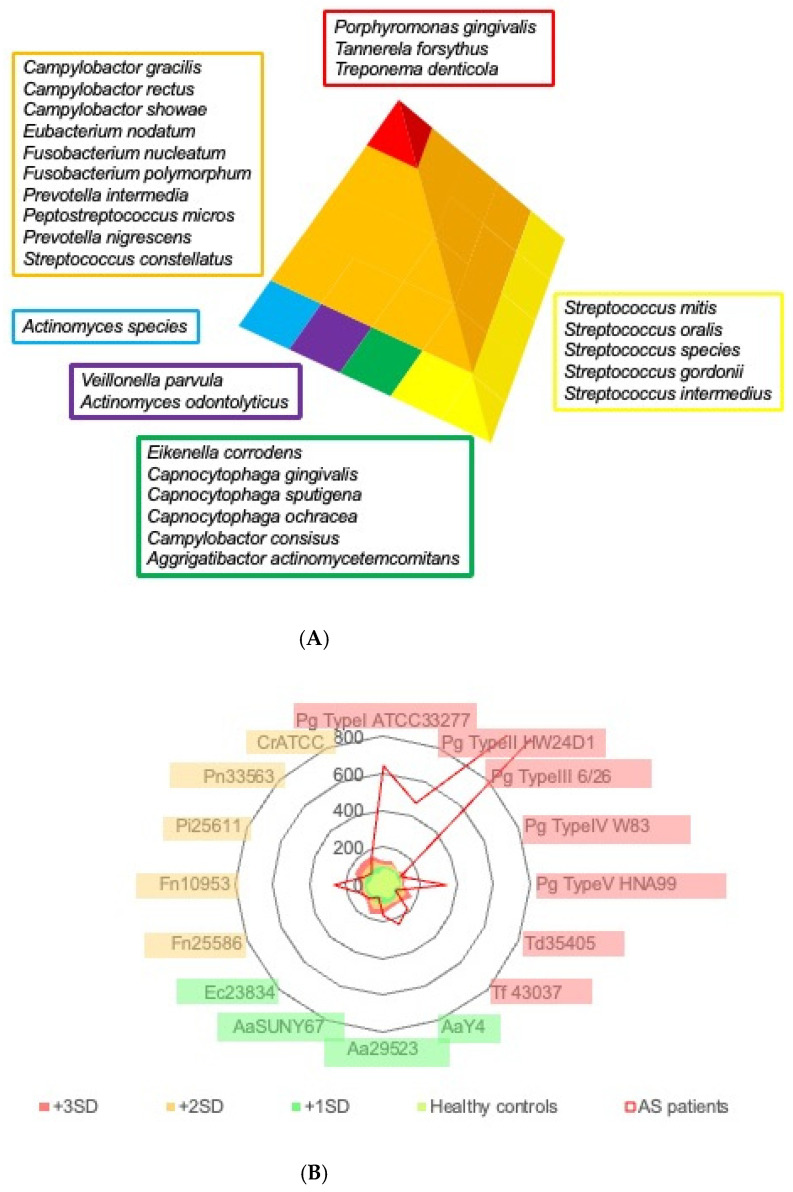
Status of antibody titer to periodontal pathogenic bacteria. (**A**) Schematic illustration of periodontal pathogenic bacteria classified by Socransky’s criteria. Periodontal pathogenic bacteria exist in a defined order. They are divided into six categories, and each is coded by color. Blue, purple, green, and yellow complexes form the basic attachment to the body surfaces such as teeth and mucosa. The red complex is a definite pathogen in periodontitis, while the orange complex primarily functions as a connecting anchor for the members of the red complex within the biofilm. (**B**) Average antibody titer against representative periodontal pathogenic bacteria. Antibody titers against each pathogenic bacterium are standardized to the average value of healthy controls. The red area in the radar chart demonstrates a +3 S.D. titer from healthy controls, the orange area a +2 S.D., and the green area a +1 S.D., respectively. The average of AS/AR patients in this study is shown in the red line. *Porphyromonas gingivalis (Pg)*, *Treponema denticola (Td), Tannerella forsythia (Tf)*, *Aggregatibacter actinomycetemcomitans (Aa), Eikenella corrodens (Ec)*, *Fusobacterium nucleatum (Fn)*, *Prevotella intermedia (Pi)*, *Prevotella nigrescens (Pn)*, and *Campylobacter rectus (Cr)*. (**C**) Correlation of antibody titers against the periodontal pathogenic bacteria and periodontal stage of AS/AR patient. Antibody titers against bacteria classified into red, orange, blue, and green complexes were summed as an index and plotted for the 13 patients who had remaining teeth and for whom serum could be collected. The antibody titer of healthy individuals was set to 100. Lines in the figure demonstrate regression lines.

**Figure 3 microorganisms-13-01677-f003:**
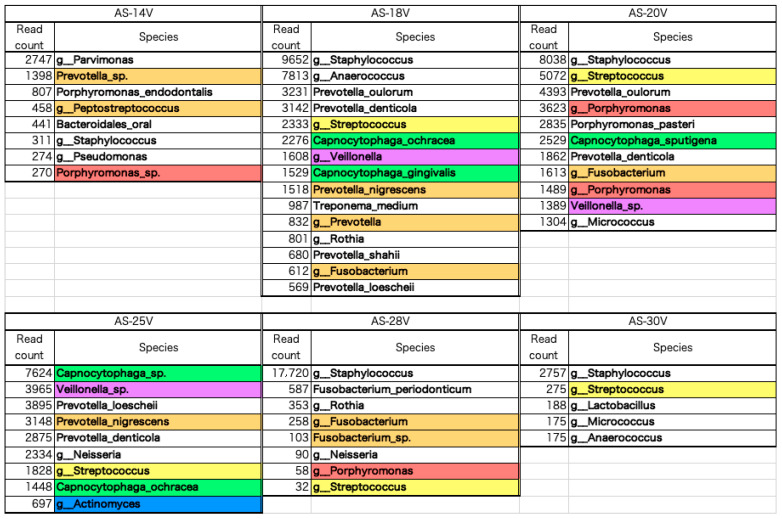
Oral commensal bacteria and periodontal pathogenic bacteria detected in the resected aortic valve. Among the bacteria detected in the resected aortic valve, oral commensal bacteria and periodontal pathogenic bacteria (color-coded as red, orange, yellow, green, blue, and purple), according to Socransky’s classification, were extracted from the lists of the top 25 bacteria (Appendix A) and are shown in Figure 3. In cases 20 and 25, red and orange bacteria were detected in the aortic valve. In case 30, neither red nor orange complex bacteria were detected in the aortic valve. In cases 14, 18, and 28, red and orange bacteria were detected in the aortic valve.

**Figure 4 microorganisms-13-01677-f004:**
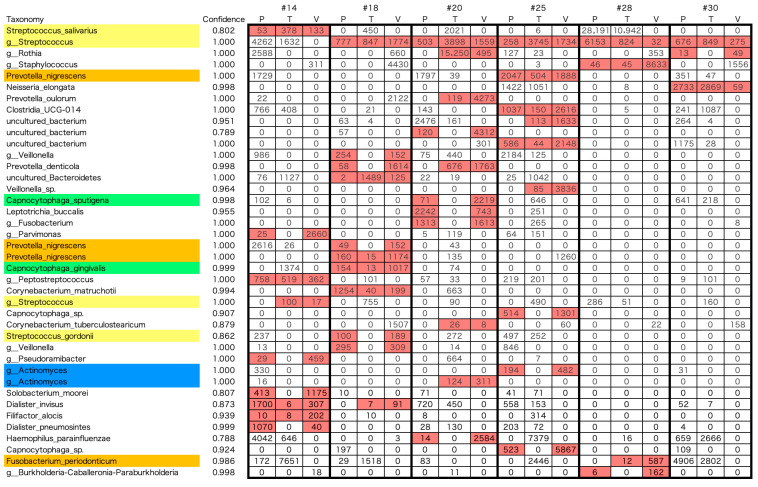
List of bacteria to which the ASV was matched in dental plaque, tongue, and valve. Matched samples are highlighted by a red background. The numbers in the tables represent the read count of each bacterium. The color of the background of the bacteria name on the left column was the same color as the bacteria matching Sokransky’s color code in Figure 2. #: case number, P: dental plaque, T: dorsal surface of tongue, V: aortic valve.

**Table 1 microorganisms-13-01677-t001:** Demographic data of the AS/AR patient.

Age		
Mean ± 1SD	74.125 ± 8.172	
Median	76	
	N	%
Gender		
male	17	53.125
female	15	46.875
Stage of periodontitis		
I	5	15.625
II	2	6.25
III	5	15.625
IV	10	31.25
edentulous	10	31.25
Number of remaining teeth		
21~	11	34.375
11~20	6	18.75
1~10	5	15.625
Valvular findings (duplicate)		
calcification	26	81.25
bicuspid	7	21.875
no abnormal findings	4	12.5
Aortic valve disease		
severe AS	25	78.125
moderate–severe AS	2	6.25
moderate AS	3	9.375
severe AR	1	3.125
moderate–severe AR	1	3.125
Number of patients with bacterial DNA detected from resected aortic valves by PCR; in total 32 patients examined	12	37.5
Number of patients with sufficient bacterial DNA collected from aortic valves for metagenomic analysis in the 12 patients examined	6	50
Number of patients with 100% ASV match between oral bacteria and bacteria collected from aortic valves in the 6 patients examined	6	100

## Data Availability

The original contributions presented in this study are included in the article/Appendix A. Further inquiries can be directed to the corresponding authors.

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
