# Peer review of "Oral Dysbiosis Is Associated with the Pathogenesis of Aortic Valve Diseases"

_microorganisms, 2025, doi:10.3390/microorganisms13071677_

Round 1
Reviewer 1 Report
Comments and Suggestions for Authors
This is an interesting and well-written manuscript titled “Oral Dysbiosis and Resulting Bacteremia Are Associated with Aortic Valve Diseases.” The manuscript contains sufficient information and presents a relevant topic that will be of interest to readers.
Material and methods section. Please make sure that all reagents and equipment mentioned are accompanied by the corresponding brand, model, and country of origin, where applicable.
Results section. It is recommended to include a summary table of the clinical data, instead of displaying individual patient information, to enhance clarity and readability
Line 250: replace the term 'bacterial flora' with 'bacterial microbiota. Throughout the text, you may consider replacing this term; alternatives such as 'bacterial microbiota', 'microbiome', or 'oral microbiota' can be used, depending on the context.
Conclusion section. It is recommended to include a brief paragraph discussing the clinical relevance of this study.
Author Response
- This is an interesting and well-written manuscript titled “Oral Dysbiosis and Resulting Bacteremia Are Associated with Aortic Valve Diseases.”The manuscript contains sufficient information and presents a relevant topic that will be of interest to readers.
A: Thank you very much for your warm comments on our manuscript.
- Material and methods section. Please make sure that all reagents and equipment mentioned are accompanied by the corresponding brand, model, and country of origin, where applicable.
A: Yes, we added the details for all reagents and equipment.
Section 2.2. The serum antibody test against periodontal disease-causing bacteria is not commercially available, but it is the original method developed by our collaborators, as described in reference 10. Although we did not think it necessary to describe the details in this manuscript, we have included the method's details.
Section 2.3. All information on the kits used for bacterial DNA extraction is provided in the text.
Section 2.4. All items required for the experiments performed are included.
Section 2.5. All items required for the experiments performed are included. We added a reference (#12).
Section 2.6. We corrected some errors in the references of the original manuscript (#13-15). Additionally, references to the methods used for alpha and beta diversity analysis and PCoA were included in section 2.7 (#17-21).
- Results section. It is recommended to include a summary table of the clinical data, instead of displaying individual patient information, to enhance clarity and readability
A: Thanks for the great suggestion. I added a new Table 1 and made the old Table 1 as supplement data (new Table S1).
- Line 250: replace the term 'bacterial flora' with 'bacterial microbiota. Throughout the text, you may consider replacing this term; alternatives such as 'bacterial microbiota', 'microbiome', or 'oral microbiota' can be used, depending on the context.
A: We changed ‘bacterial flora’ to ‘bacterial microbiota’. We changed ‘bacterial flora’ and ‘flora’ to ‘bacterial microbiota’ throughout the manuscript.
- Conclusion section. It is recommended to include a brief paragraph discussing the clinical relevance of this study.
A: Thanks for the great suggestion. However, as pointed out by other reviewers, the results of this study do not allow us to conclude that the presence of oral bacteria in the aortic valve is a cause or a progression factor of aortic valve disease. We slightly modified the Conclusion.
Reviewer 2 Report
Comments and Suggestions for Authors
Major Comments
The conclusions imply a causal link between oral dysbiosis and aortic valve disease. However, while the detection of identical ASVs is suggestive of translocation, it does not establish causality. Please revise the wording in the abstract, results, and discussion to express a more cautious interpretation.
Only six aortic valve samples yielded suitable DNA for sequencing. This limitation should be discussed more explicitly in the manuscript, especially in terms of statistical power and generalizability.
The absence of healthy controls or patients without aortic disease reduces the ability to attribute the findings to pathology. Authors should justify this omission and address how it affects interpretation.
Some samples required repeated PCR amplification. The risk of nonspecific amplification or contamination must be addressed more clearly. What quality control measures were in place?
The term "oral dysbiosis" is frequently used but not operationally defined. Please specify the criteria used to determine dysbiosis in the patient population (e.g., diversity indices, pathogen load).
Minor Comments
The manuscript would benefit from professional language editing. Numerous grammatical issues and awkward sentence structures detract from clarity and flow.
Figures 3 and 4, along with some tables, are overly complex and not fully interpretable on their own. Please enhance legends, consider breaking large tables into simpler components, and highlight key comparisons.
Include more recent studies on the systemic effects of oral microbiota and microbial translocation to provide a stronger context for the findings.
Clarify the thresholds used to determine that ASVs were "identical" across tissues. Was a 100% identity required? How were ambiguous or near-matches handled?
The term "oral dysbiosis" is frequently used but never defined. At times, it seems to refer to microbial imbalance, and at others to disease state. This should be clearly defined and used consistently.
The phrase "hierarchy of bacterial species" is not conventional. "Composition," "distribution," or "relative abundance" would be more scientifically accurate terms in this context.
Use of "flora" and "microbiota" interchangeably. Please use the more current and specific term "microbiota" when referring to microbial communities.
Comments on the Quality of English LanguageThe manuscript is generally understandable; however, the quality of English needs improvement to ensure clear and professional communication of the research. There are many grammatical errors, awkward sentences, and inconsistent terms that make the text harder to read. It is highly recommended to have a professional scientific English editor review the manuscript to improve its quality and impact.
Author Response
- The conclusions imply a causal link between oral dysbiosis and aortic valve disease. However, while the detection of identical ASVs is suggestive of translocation, it does not establish causality. Please revise the wording in the abstract, results, and discussion to express a more cautious interpretation.
A: As pointed out, our data only show that clones identical to the oral bacteria were present in the aortic valve. As background, our experiment also shows that patients with aortic valve replacement suffer from more severe periodontal disease than the general population. These data obtained from clinical observational studies suggest that oral bacteria are likely to be associated with the pathogenesis of aortic valve diseases. Further studies, perhaps animal studies, are needed to show whether oral bacteria are responsible for the onset or progression of aortic valve diseases. The Discussion was modified and the Conclusions were also modified.
- Only six aortic valve samples yielded suitable DNA for sequencing. This limitation should be discussed more explicitly in the manuscript, especially in terms of statistical power and generalizability.
A: The data presented in this study, which shows the detection of oral bacteria in the aortic valve, have only been published in a few papers in the past, and the data showing that the oral bacterial clone and the clone present in the aortic valve were identical seems to be the first report. That is the topic of this paper, and I do not think it is possible to discuss whether the results can be generalized by statistical analysis how many of the patients with aortic valve disease had bacteria on the resected aortic valves, how many of them contained oral bacteria, and in how many of those patients the same clone of bacteria was present. This study was a clinical observational study to examine the possibility of an association between aortic valve disease and oral bacteria by identifying patients in whom the same clone of oral bacteria was present in the resected aortic valves due to aortic valve disease. As the culmination of our 40 years of our experience in molecular biological research, DNA extraction and PCR were performed in a sophisticated manner with high sensitivity and high specificity. Bacterial DNA could be detected in aortic valves in only 12 of the 32 cases, and sufficient bacterial DNA could be extracted for metagenomic analysis in only 6 cases, this is a fact. However, someone raises the question of whether this is a fact or a matter of manipulation. We added to the discussion that this could be a potential limitation in the interpretation of the results.
- The absence of healthy controls or patients without aortic disease reduces the ability to attribute the findings to pathology. Authors should justify this omission and address how it affects interpretation.
A: We added a description concerning this issue in the Discussion. It is not possible to histologically observe aortic valves in healthy individuals or in patients with mild valvular disease and extract bacterial DNA for metagenomic analysis.
- Some samples required repeated PCR amplification. The risk of nonspecific amplification or contamination must be addressed more clearly. What quality control measures were in place?
A: Molecular biology experiments in general are performed in a sophisticated manner by 40 years of our experience and are not expected to cause problems that would confound the interpretation of the results. As noted in the text, we understand that it is problematic to quantitatively discuss these three cases, for which a library of bacterial DNA could not be constructed without performing a second round of PCR. In fact, the beta diversity in these cases differs from that in other cases, and many unassigned fragments appeared, which may indicate nonspecific amplification. However, examining the clones identified in these three cases reveals no evidence of contamination between samples, and the 18S RNAs that were thought to be specifically amplified can be compared qualitatively. In these three cases, the bacteria detected in the aortic valve exhibited the same ASV as the oral bacteria, indicating that they might be the same clone in each case. We added a note to the Discussion as a limitation.
- The term "oral dysbiosis" is frequently used but not operationally defined. Please specify the criteria used to determine dysbiosis in the patient population (e.g., diversity indices, pathogen load).
A: We have added an explanation of how we intend to use the term “oral dysbiosis” in the abstract and the introduction.
In this paper, we evaluated the increase of Red Complex in oral bacteria, the increase of serum antibodies against Red Complex, the progression of periodontal disease, and tooth loss as outcomes of “oral dysbiosis”. We did not compare the alpha or beta diversity of oral bacteria between patients and healthy controls. In addition, some patients had a higher frequency of detection of red complex as a pathogen of periodontal disease, but this was also not compared with healthy subjects.
- The manuscript would benefit from professional language editing. Numerous grammatical issues and awkward sentence structures detract from clarity and flow.
A: A native English speaker edited the original manuscript before submission. We checked our English grammar, spelling and punctuation as carefully as possible in several ways. This is the best we can do.
- Figures 3 and 4, along with some tables, are overly complex and not fully interpretable on their own. Please enhance legends, consider breaking large tables into simpler components, and highlight key comparisons.
A: Fig 3 shows the top 25 bacterial number (read counts) detected in the tongue, periodontal pocket, and resected aortic valve in each case. This is clearly indicated by color-coding, which bacteria are present in which sites according to Socransky's functional classification of oral bacteria (described in Figure 2). The figure might be confusing because only four types of Sokransky's color codes are listed in the left column. We removed the left column and added the figure legend to explain that the bacteria matching each of the color codes in Figure 2 were displayed in the same color.
Fig. 4 shows the frequency of ASV matches among the bacteria detected at the three locations. The color code for the bacteria name in the left column is Sokransky's color code in Figure 2. We slightly modified the legend.
- Include more recent studies on the systemic effects of oral microbiota and microbial translocation to provide a stronger context for the findings.
A: Yes, we added the recent literature (new #27-29, 34-38).
- Clarify the thresholds used to determine that ASVs were "identical" across tissues. Was a 100% identity required? How were ambiguous or near-matches handled?
A: We picked up a 100% match for the V3-V4 region of the 16S rRNA sequence. We added this information to the text.
- The term "oral dysbiosis" is frequently used but never defined. At times, it seems to refer to microbial imbalance, and at others to disease state. This should be clearly defined and used consistently.
A: The answer is the same as the answer to Q5.
- The phrase "hierarchy of bacterial species" is not conventional. "Composition," "distribution," or "relative abundance" would be more scientifically accurate terms in this context.
A: We changed it to “distribution”.
- Use of "flora" and "microbiota" interchangeably. Please use the more current and specific term "microbiota" when referring to microbial communities.
A: We changed it to “microbiota”.
- Comments on the Quality of English Language. The manuscript is generally understandable; however, the quality of English needs improvement to ensure clear and professional communication of the research. There are many grammatical errors, awkward sentences, and inconsistent terms that make the text harder to read. It is highly recommended to have a professional scientific English editor review the manuscript to improve its quality and impact.
A: The answer is the same as the answer to Q5.
Reviewer 3 Report
Comments and Suggestions for Authors
This manuscript investigates the association between oral dysbiosis and aortic valve disease using metagenomic analysis of bacterial DNA from excised aortic valves and oral specimens (dental plaque and tongue swabs). While the manuscript presents interesting findings, several major and minor issues in study design, methodology, interpretation, and presentation must be addressed before it is suitable for publication.
In the materials and methods section, I have many remarks. The inclusion criteria specify patients undergoing valve replacement for AS or AR, but there is no clarity on exclusion criteria, comorbid conditions (e.g., diabetes, immunosuppression), or prior antibiotic use, which may significantly impact the oral and valve microbiota.
A significant limitation is the lack of a control group of patients without aortic valve disease. Comparing microbial profiles and antibody titers to a healthy surgical cohort (e.g., CABG patients) would substantially strengthen the inference of a causal link between oral dysbiosis and valve disease. I suggest for the authors to clarify and justify patient selection, and consider inclusion of a control cohort.
Only 6 out of 32 valve samples yielded sufficient DNA for sequencing. This low yield raises concerns about sampling adequacy, potential technical bias, and generalizability. Also, the authors acknowledge that some samples required amplification of PCR products from a previous PCR, which increases the risk of contamination and chimeric sequences. These limitations should be explicitly discussed in the Discussion section.
Although the study used 16S rRNA amplicon sequencing and ASV-based identification, the paper does not report confidence scores or bootstrap values for species-level assignments. The identification of “identical ASVs” in oral and valve samples is central to the paper’s claim but needs further validation. Consider performing phylogenetic trees or using full-length 16S or metagenomic sequencing to strengthen this evidence.
The correlation between antibody titers and periodontal disease stage is informative. However, the selection of bacterial strains for ELISA and the controls (presumably healthy adults) need more methodological description. Were the controls age- and sex-matched? Also, it’s unclear if the ELISA method was validated for specificity and cross-reactivity, especially for closely related species in the same complexes. I suggest to the authors to provide more technical detail on ELISA, DNA extraction, and sequencing quality metrics.
In the Results section, the assertion that oral bacteria colonize the valve and contribute to pathology is speculative, especially given the cross-sectional nature of the study. Alternative interpretations (e.g., transient bacteremia leading to incidental colonization) should be discussed.
Authors should strengthen the Discussion section on alternative hypotheses, potential contaminants, and the limitations of 16S rRNA sequencing. Consider rephrasing claims regarding causality to better reflect the data presented.
In the Conclusion that oral dysbiosis “may be associated with the onset or progression” of AV disease requires careful qualification. Consider tempering these claims or supporting them with stronger longitudinal or mechanistic data.
Some minor suggestions:
The abstract is clear but should mention the small number of valve samples successfully sequenced to temper expectations. Figures S1–S3 are informative but could benefit from improved resolution and clearer legends. Some taxa are unreadable due to font size or overlapping labels. Figure 3 (taxa ranked by read count) lacks statistical comparisons. Consider using bar plots or heatmaps with significance indicators. Please improve figure clarity and ensure that all taxa are legible.
In the statistical analisis the alpha and beta diversity metrics are appropriately used, but details on correction for multiple comparisons (e.g., Bonferroni or FDR) are not reported. And PERMANOVA results (p = 0.001) support differences among groups, but effect sizes (e.g., R² values) should be included.
The manuscript contains several grammatical errors and awkward phrasings (e.g., “flora may be diverse” → “the bacterial community may be diverse”).
Editing for clarity and conciseness is recommended throughout.
Author Response
- This manuscript investigates the association between oral dysbiosis and aortic valve disease using metagenomic analysis of bacterial DNA from excised aortic valves and oral specimens (dental plaque and tongue swabs). While the manuscript presents interesting findings, several major and minor issues in study design, methodology, interpretation, and presentation must be addressed before it is suitable for publication.
A: Thank you; I corrected the manuscript as follows, according to your suggestions.
- In the materials and methods section, I have many remarks. The inclusion criteria specify patients undergoing valve replacement for AS or AR, but there is no clarity on exclusion criteria, comorbid conditions (e.g., diabetes, immunosuppression), or prior antibiotic use, which may significantly impact the oral and valve microbiota.
A: Thank you for the excellent suggestions. The inclusion criteria for this study were that the Aortic valves had been resected during the specified period and the material was aseptically available. No exclusion criteria were defined. We did check for DM, immunosuppression, and preoperative antibiotics administration, and described them in the Materials and Methods.
- A significant limitation is the lack of a control group of patients without aortic valve disease. Comparing microbial profiles and antibody titers to a healthy surgical cohort (e.g., CABG patients) would substantially strengthen the inference of a causal link between oral dysbiosis and valve disease. I suggest for the authors to clarify and justify patient selection, and consider inclusion of a control cohort.
A: As pointed out, no controls were put in this study for comparison. It is not possible to analyze the aortic valve in normal subjects. We are currently planning to analyze tissues from patients with other cardiovascular diseases, but as a preliminary experiment, we were unable to obtain sufficient tissue from an atherosclerotic artery. This is an observational study of patients who underwent aortic valve replacement and had the valve aseptically obtained. The objective of the study was to determine whether bacteria could be detected in the aortic valve and whether the detected bacteria matched those found in the oral cavity. Numerous cohort studies have investigated the relationship between periodontal disease and various vascular and cardiac diseases. Based on the results of these studies, we aim to clarify the direct association of oral bacteria in the pathogenesis of aortic valve disease. Please understand the aim of our study.
The comparison of periodontal disease severity was based on the results of a large Japanese survey. According to the suggestions of the other reviewers, Fig. 1 was slightly modified, a few descriptions were added, and the results of the statistical tests were presented.
We compared blood antibody titers with those of healthy controls in our previous searches. Information on controls was added to the Materials and Methods section.
- Only 6 out of 32 valve samples yielded sufficient DNA for sequencing. This low yield raises concerns about sampling adequacy, potential technical bias, and generalizability. Also, the authors acknowledge that some samples required amplification of PCR products from a previous PCR, which increases the risk of contamination and chimeric sequences. These limitations should be explicitly discussed in the Discussion
A: The DNA extracted from the aortic valve showed that the OD260/280 was about 1.8. However, bacterial DNA could be detected in 12 of the 32 cases. Six of these cases contained only traces of bacterial DNA, meaning that only the remaining six cases could be used for bacterial metagenomic analysis as described in the text. We believe that all our experiments are performed with a high degree of accuracy. It is rather surprising to us that bacteria were detected in aortic valves, and 12 out of 32 cases were positive, which is one of the key findings of this study.
- Although the study used 16S rRNA amplicon sequencing and ASV-based identification, the paper does not report confidence scores or bootstrap values for species-level assignments. The identification of “identical ASVs” in oral and valve samples is central to the paper’s claim but needs further validation. Consider performing phylogenetic trees or using full-length 16S or metagenomic sequencing to strengthen this evidence.
A: ASV was compared in the V3-V4 region of the 16S rRNA sequencing. The V3-V4 region of 16S rRNA is a region where the characteristics of microbial species can be observed and is widely used for estimating microbial taxonomy based on DNA sequence; therefore, this region is commonly used to construct a 16S rRNA library. It can be concluded that the oral bacteria and the bacteria detected in the aortic valve are the same clone. We slightly modified the Results.
- The correlation between antibody titers and periodontal disease stage is informative. However, the selection of bacterial strains for ELISA and the controls (presumably healthy adults) need more methodological description. Were the controls age- and sex-matched? Also, it’s unclear if the ELISA method was validated for specificity and cross-reactivity, especially for closely related species in the same complexes. I suggest to the authors to provide more technical detail on ELISA, DNA extraction, and sequencing quality metrics.
A: The sera from five healthy subjects (ages 12–81) were pooled and used for calibration. We added the ELISA method because we did not describe the details of the method, only citing our previous papers. ELISA is a specific test used in real clinical examinations. DNA extraction, library preparation for metagenomic analysis, and NGS sequencing were performed as described in the manuscript, and no special methods were used.
- In the Results section, the assertion that oral bacteria colonize the valve and contribute to pathology is speculative, especially given the cross-sectional nature of the study. Alternative interpretations (e.g., transient bacteremia leading to incidental colonization) should be discussed.
A:As pointed out, this is a cross-sectional clinical observational study. Then, it is not possible to prove the causative involvement of the bacteria present in the development or progression of aortic valve disease. In fact, we were able to demonstrate that the same bacterial clones present in the oral cavity were also present in the resected aortic valves from patients with aortic valve disease. Based on this fact, we can infer that oral bacteria play a role in the pathogenesis of aortic valve disease. It is impossible to obtain further evidence from human studies using clinical materials; therefore, we must rely on animal studies. But even if we have a good animal model and it is proven, we cannot say that the same thing happens in humans.
Transient bacteremia and instantaneous adhesion of oral bacteria to the aortic valve cannot be completely ruled out. However, all patients received intravenous antibiotics during the perioperative period, and the aortic valve was physically impacted at the time of resection and washed with saline. Therefore, it is unlikely that transient adhesion of extremely small amounts of circulating bacteria could be detected. I have modified the descriptions in the Results, Discussion, and Conclusion sections.
- Authors should strengthen the Discussion section on alternative hypotheses, potential contaminants, and the limitations of 16S rRNA sequencing. Consider rephrasing claims regarding causality to better reflect the data presented.
A:  In response to the opinion of another reviewer, it is unlikely that there is bacterial contamination in the process of analysis. In addition, based on our 40 years of experience in molecular biological research and the results of analysis, it is unlikely that there is any contamination among the samples. We added the description of a potential limitation of the experiments using PCR in general.
- In the Conclusion that oral dysbiosis “may be associated with the onset or progression” of AV disease requires careful qualification. Consider tempering these claims or supporting them with stronger longitudinal or mechanistic data.
A: As pointed out, and as we have answered the other reviewers, we have made some modifications.
- The abstract is clear but should mention the small number of valve samples successfully sequenced to temper expectations. Figures S1–S3 are informative but could benefit from improved resolution and clearer legends. Some taxa are unreadable due to font size or overlapping labels. Figure 3 (taxa ranked by read count) lacks statistical comparisons. Consider using bar plots or heatmaps with significance indicators. Please improve figure clarity and ensure that all taxa are legible.
A: As pointed out by other reviewers, we modified the description.
FigS1-3 was modified, and the font was enlarged.
Considering the purpose of this study, it would not make sense to statistically compare the read counts in Fig. 3 between patients or by site within the same patient. The names and frequencies of all bacteria detected in each patient and tissue are listed in Fig. S1. A new heat map was created for all bacteria detected (new Fig. S2). There are no findings beyond those described in the results in the original manuscript, and therefore, no new additional statements are made as interpretable results. As instructed by the other reviewer, a heat map by bacterial phylum was created in accordance with the reviewer's suggestion as a new Fig. S4.
- In the statistical analisis the alpha and beta diversity metrics are appropriately used, but details on correction for multiple comparisons (e.g., Bonferroni or FDR) are not reported. And PERMANOVA results (p = 0.001) support differences among groups, but effect sizes (e.g., R² values) should be included.
A: The analysis was performed according to the reviewer's instructions and is included as a new Table S2, S3, and S4. The descriptions and interpretations of each were added to the Materials and Methods and Results sections.
- The manuscript contains several grammatical errors and awkward phrasings (e.g., “flora may be diverse” → “the bacterial community may be diverse”).
A: Other reviewers have also pointed out this point; we have corrected it to “bacterial microbiota”.
A native English speaker edited the original manuscript before submission. We checked our English grammar, spelling and punctuation as carefully as possible in several ways. This is the best we can do.
Reviewer 4 Report
Comments and Suggestions for Authors
Comments on Yaguchi et al.
This research attempted to compare the bacterial composition of resected aortic valve of patients with aortic stenosis and aortic regurgitation with the oral microbiome. According to the abstract, the authors state that they observed high similarity in bacterial composition on the aortic valve as in the oral cavity suggesting that bacteria from the oral cavity can infect the aortic valve. They also claim that patients with aortic valve disease have severe periodontal disease. But the data presented do not show this tendency. Only selected samples are compared. The text needs more descriptions and explanations. The specific bacterial names should be mentioned, not sufficient to write numbers of species. An integrative analysis should be better presented for clarity.
Abstract:
Line 37: If you use the concept "more" you need to state in comparison to what? Control?
Line 39: "sometimes" should be expressed in number. What about other patients?
The kind of oral dysbiosis should be stated in the abstract.
Introduction:
There are too many times of "the" in the text. Please remove unnecessary ones.
Line 53: Please verify the sentence: "Aortic valve disease (AV) is a known cause of IE". Bacterial infection may cause IE, but I am not sure if AV causes IE, or only increases the risk of IE.
Line 65: You need to specify exactly what you observed. It is not logical when you write first "identical bacterial species", but thereafter you write "only in one case". It is unclear how this can be.
Line 69: "attempted" is not a scientific word. Please rephrase.
The introduction should include more data from the literature.
The novelty of the study should be stated.
Methods:
Did you have a control group or comparison? In the result section it states 2378 patients – then the comparison is done with healthy or ill patients of the same age? This has to be stated in the method section.
All methods should be described in detail, and how the analyses were made.
Section 2.2: The source of the ELISA should be stated. It is not sufficient to refer to previous studies. The exact procedure should be described. Why were different DNA extraction methods used for the aortic valves and the samples from the oral cavity.
Line 125 and 126: You need to define what the H, R, B and V letters stand for.
Section 2.6 lacks a title.
The method section lacks a statistic section.
Results:
In general, the text needs to provide more explanations of the graphs and how you came to these data.
Line 159: The numbers should be stated together with std.
Line 160: I think there is a typographic error: "severe" is added in all groups. Please correct.
You need to define how you classified periodontitis into the 4 stages.
Figure 2B. You need to define the different bacterial abbreviations in the legend. The presentation is difficult to understand. I think it would be better to present the data in bar graphs, with individual dots of the AS and AR patients.
Figure 2C should have a title of the Y-axis. Line 237: spelling mistake, correct to "titers". You need to explain the lines in the legend.
In Figure legend 2 you need to state how many samples were used for the plots.
Figure S1: It seems that the labels are not exactly below the bar. You need to state the color of the most prevalent bacteria, or point to the population with an arrow, since the color list is quite unreadable. This list needs to be made readable. Some statistics are required. The legend should have a space between "have" and "caused". When I am trying to compare between the three bars of each individual, it seems that the bacterial composition is not "identical" as stated in the text. All samples are quite different. So, the text should be corrected.
For instance, samples AS-14V, AS-28V and AS-30V have high similarities, but not with their respective P and T samples.
What about the T and P data of the AS patients without bacteria on the aortic valve?
I would suggest making an addition graph to compare between the individual bacterial phylum.
Section 3.3 should have much more comprehensive text, with many more comparisons.
It seems not logical that the aortic valves are infected with so many bacteria. How could you eliminate any contamination during isolation?
Figure 3: You need to describe the meaning of the numbers.
Something is lacking in the text of the Result section. A deeper analysis is required, and the text needs to provide more concrete data. Some more conclusive data are required. All text is description of individual samples, but there is no domination or trend.
The different nomenclatures, e.g. ASV should be defined. (amplicon sequence variant).
Figure S3 shows that AS differs from Plaque and tongue. So, the word "identical" repeatedly used in the text is not accurate.
And if you find "identical", the specific species should be stated.
The discussion should state the bacterial names. The text is too superficial and not conclusive.
The conclusion should not be an estimate, but a conclusion from the data obtained from this study.
Comments on the Quality of English LanguageThe scientific writing needs to be improved.
Author Response
- This research attempted to compare the bacterial composition of resected aortic valve of patients with aortic stenosis and aortic regurgitation with the oral microbiome. According to the abstract, the authors state that they observed high similarity in bacterial composition on the aortic valve as in the oral cavity suggesting that bacteria from the oral cavity can infect the aortic valve. They also claim that patients with aortic valve disease have severe periodontal disease. But the data presented do not show this tendency. Only selected samples are compared. The text needs more descriptions and explanations. The specific bacterial names should be mentioned, not sufficient to write numbers of species. An integrative analysis should be better presented for clarity.
A: Thank you for your critical comment. We followed the reviewers' suggestions and revised the manuscript as positively as possible. Although it is not possible to directly prove a causal relationship because this study is a clinical observation, we believe that our conclusions can be drawn by combining the facts obtained.
The names of bacterial genus and species shown in Fig. 3, Fig. 4, and Fig. S1 are deduced from the 16S rRNA sequence (>70% confidence threshold) and only mean that the bacteria with the indicated names are possible. In Fig. 4, several bacteria with perfectly matched amplicon sequences between the oral bacteria and those detected in the aortic valve are shown, which appears to be direct proof of identity.
Although it is criticized for its analysis of selected samples, no previous report has examined bacterial identity using metagenomic analysis or amplicon sequencing of oral bacteria and those detected in the aortic valve. As described in the manuscript, only 6 samples can be analyzed for metagenomics by the fact, not by selection, because only 6 of the 32 cases had sufficient amounts of bacterial DNA extracted from the aortic valve for metagenomic analysis. The abstract was slightly changed.
- Line 37: If you use the concept "more" you need to state in comparison to what? Control?
A: We changed the description, Fig. 1, and the legends.
- Line 39: "sometimes" should be expressed in number. What about other patients?
A: We changed the description (l39-41).
- The kind of oral dysbiosis should be stated in the abstract.
A: We added an explanation of how we intend to use the term “oral dysbiosis” in the abstract (l42-44) and the introduction.
- There are too many times of "the" in the text. Please remove unnecessary ones.
A: A native English speaker edited the original manuscript before submission. We checked our English grammar, spelling and punctuation as carefully as possible in several ways. This is the best we can do.
- Line 53: Please verify the sentence: "Aortic valve disease (AV) is a known cause of IE". Bacterial infection may cause IE, but I am not sure if AV causes IE, or only increases the risk of IE.
A: In our understanding of the pathogenesis of IE, the blood flow jets caused by the valvular diseases (regurgitation and stenosis) damage the valves and endocardium, resulting in the formation of a sterile vegetation at the site of the damage. Bacterial infection on these vegetations is thought to lead to the development of IE. We slightly modified the description as “Aortic valve disease (AV) is known to increase the risk of IE.”
- Line 65: You need to specify exactly what you observed. It is not logical when you write first "identical bacterial species", but thereafter you write "only in one case". It is unclear how this can be.
A: As described in the text, in our previous study, bacterial cultures of blood from patients with IE were compared with dental plaque cultures to verify the identity of the oral bacteria and the causative bacteria of IE. The culture test was able to detect identical bacterial species, which were identified morphologically and biochemically in several cases. However, in only one case, the bacteria detected from dental plaque and blood had a complete match in the 16S rRNA gene sequence. There is no inconsistency in the description. We slightly modified the description.
- Line 69: "attempted" is not a scientific word. Please rephrase.
A: We deleted “attempted”.
- The introduction should include more data from the literature.
A: To the best of our knowledge, there have been no reports of the detection of bacteria from aortic valves by metagenomic analysis to prove their identity with oral bacteria. We cited additional papers (new references #27-29, #34-38) on the association of oral bacteria with the detection of bacteria from other organs, including the aortic valve.
- The novelty of the study should be stated.
A: All the results summarized in the first paragraph of the discussion in the original manuscript are novel. The implications and conclusions drawn from them are also novel. We slightly modified the description to emphasize the novelty.
- Did you have a control group or comparison? In the result section it states 2378 patients – then the comparison is done with healthy or ill patients of the same age? This has to be stated in the method section.
A: As we answered the Q from reviewer 1, the comparison of periodontal disease severity was based on the results of a large Japanese survey. According to the suggestions of the other reviewers, Fig. 1 was slightly modified, a few descriptions were added, and the results of the statistical tests were presented.
- All methods should be described in detail, and how the analyses were made.
A: Methods for all wet and dry experiments are described in detail except for the method in section 2.2 of the next question.
- Section 2.2: The source of the ELISA should be stated. It is not sufficient to refer to previous studies. The exact procedure should be described. Why were different DNA extraction methods used for the aortic valves and the samples from the oral cavity.
A: Section 2.2. The serum antibody test against periodontal disease-causing bacteria is not commercially available; however, it is the original method developed by our collaborators, as described in reference 10. Although we did not think it necessary to describe the details in this manuscript, we have included the method's details.
Regarding DNA extraction, the process of extracting DNA from microorganisms embedded in tissue differs from extracting DNA from the bacteria themselves.
- Line 125 and 126: You need to define what the H, R, B and V letters stand for.
A: I believe that the notation of mixed bases is an international standard, so there is no need to explain it.
Symbol |
R |
M |
W |
S |
Y |
K |
H |
B |
D |
V |
N |
Nucleic acids |
A,g |
A,C |
A,T |
C,g |
C,T |
g,T |
A,T,C |
g,T,C |
g,A,T |
A,C,g |
A,C,g,T |
- section 2.6 lacks a title.
A: We added. “Microbial population analysis”.
- The method section lacks a statistic section.
A: We added a statistics section.
- In general, the text needs to provide more explanations of the graphs and how you came to these data.
A: The old Table 1 was renamed Supplemental Table 1, and a new Table 1 was created. Basic patient information was organized, and a description of it was added to the text.
The results of the statistical test are presented in Figure 1, along with an explanation in the text.
New Tables S1, S2, and S3 are added and explained in the text.
Figures 2, S1, S2, 3, and 4 are all properly explained in the text.
- Line 159: The numbers should be stated together with std.
A: Mean and SD are listed in the new Table 1, and the text has been revised.
- Line 160: I think there is a typographic error: "severe" is added in all groups. Please correct.
A: Yes, we created a new Table 1 and corrected the text.
- You need to define how you classified periodontitis into the 4 stages.
A: Yes, we have added the definition of grading periodontitis in the Materials and Methods section.
- Figure 2B. You need to define the different bacterial abbreviations in the legend. The presentation is difficult to understand. I think it would be better to present the data in bar graphs, with individual dots of the AS and AR patients.
A: This radar chart is the usual way of displaying antibody titers against periodontal disease bacteria. We added the explanation of the bacterial abbreviations in the legend.
- 2C should have a title of the Y-axis.
A: Yes, we added a title of the Y-axis.
- Line 237: spelling mistake, correct to "titers". You need to explain the lines in the legend.
A: Yes, we corrected the misspelling. We also added the explanation of the line.
- In Figure legend 2 you need to state how many samples were used for the plots.
A: We added the sample number in the Fig. 2C legend. “Antibody titers against bacteria classified into red, orange, blue, and green complexes were summed as an index and plotted for the 13 patients who had remaining teeth and for whom serum could be collected.”.
- Figure S1: It seems that the labels are not exactly below the bar. You need to state the color of the most prevalent bacteria, or point to the population with an arrow, since the color list is quite unreadable. This list needs to be made readable. Some statistics are required.
A: Yes, we corrected the labels below the bar, and we added the color of the column for the most prevalent bacteria. We have included all bacteria detected by the metagenome analysis in this Figure; therefore, the Figure becomes quite complicated, as the reviewer mentioned. We enlarged the name of the bacteria and the color of the column. We only show the names and frequencies of all bacteria detected at each site for each case. Statistical analyses of detection frequencies are not applicable here, as alpha and beta diversity are being examined.
- The legend should have a space between "have" and "caused". When I am trying to compare between the three bars of each individual, it seems that the bacterial composition is not "identical" as stated in the text. All samples are quite different. So, the text should be corrected.
A: In the Fig S1 legend of our submitted manuscript, there is a space for “have” and “caused”.
In Results Section 3.3, we explained Fig. S1, but we did not say “the bacterial composition is identical”. Figure S1 shows the genus of all bacteria obtained from the metagenomic analysis. As noted in the text, bacterial genera are determined based on ASVs that are at least 70% identical. Fig. 3 shows the family, genus, and species of the top 25 ASVs detected in each patient. Fig. 3A shows bacteria detected from the aortic valve, Fig. 3 B from plaque in periodontal pockets, and Fig. 3C from plaque on the tongue, in order of frequency of detection. The bacteria detected at each site in each patient showed considerable agreement at the family and genus levels, with some cases consistent with the species level. The bacterial genera and species highlighted in red in Fig. 4 indicate the read counts of bacteria with 100% matching ASVs. This allows us to conclude that the oral bacteria and the bacteria detected from the aortic valve belong to the same clone.
- For instance, samples AS-14V, AS-28V and AS-30V have high similarities, but not with their respective P and T samples.
A: In AS-14V, AS-28V, and AS-30V, we could not obtain enough DNA for library preparation by 1st PCR, so we performed a 2nd round of PCR to prepare libraries. Therefore, these samples exhibit different β-diversity compared to other samples. This is clearly stated in Section 3.4 and in the legend in Fig. S1. Therefore, the diversity of AS-14V, AS-28V, and AS-30V is different from that of the respective oral bacteria. The purpose of this study was to identify the bacterial species and clones detected in the aortic valves, and these samples were included in the analysis as described in Results section 3.4 in the text.
- What about the T and P data of the AS patients without bacteria on the aortic valve?
A: Although we have pooled the oral bacteria in patients in whom bacterial DNA was not detected on the aortic valve, we have not done metagenomic analysis in this experiment.
- I would suggest making an addition graph to compare between the individual bacterial phylum.
A: As instructed by the reviewer, heatmaps by bacterial phylum were created and added as supplement data (Fig S3, alpha diversity and beta diversity were changed to S4 and S5, respectively). We added a note to the results. 
- Section 3.3 should have much more comprehensive text, with many more comparisons.
A:A heat map for all bacteria detected was created and added as a new figure (Fig. S2).  As for the results, as described in the original manuscript, bacteria were detectable in the aortic valves, and the oral bacteria and those detected in the aortic valves generally followed different patterns. However, some were of the same genus and species. Following the reviewer's suggestion, a heat map of the bacterial phylum was created, and the results are described above.
- It seems not logical that the aortic valves are infected with so many bacteria. How could you eliminate any contamination during isolation?
A: We were more surprised than the reviewers when we saw this result. But this is the fact. The aortic valve was aseptically removed at the time of surgery, placed in a sterile tube, and stored at -80 C until analysis. Afterwards, great care is taken to avoid contamination between samples.
- Figure 3: You need to describe the meaning of the numbers.
A: We added “read count” to the Figure.
- Something is lacking in the text of the Result section. A deeper analysis is required, and the text needs to provide more concrete data. Some more conclusive data are required. All text is description of individual samples, but there is no domination or trend.
A: We believe that only the facts obtained should be described in “the results section”. We noted the surprising fact that bacterial DNA was detected in the aortic valve of 12 out of 32 patients. The conclusive fact is that in 6 of the 12 cases, the same bacterial clone was detected in the aortic valve and oral commensal bacteria and/or periodontal pockets. We explain our interpretation of those facts in the Discussion. The results and discussions have been substantially modified for easier understanding, as suggested by the reviewers. 
- The different nomenclatures, e.g. ASV should be defined. (amplicon sequence variant).
A: In Section 2.6, we described ASV as an amplicon sequence variant in the original manuscript. The very general terminology, e.g., DNA, rRNA, PCR, etc., does not require explanation. References are provided for technical terms related to analytical methods.
- Figure S3 shows that AV differs from Plaque and tongue. So, the word "identical" repeatedly used in the text is not accurate.
A: Obviously, the distribution of oral bacteria is different from that of bacteria present in AV, which is explicitly mentioned in section 3.4 of the text. The results show that some of the clones match the bacterial genus and species, and that there are identical clones with 100% matching ASVs. We have tried to distinguish between these terms and describe them clearly and understandably. We have reviewed and revised the entire manuscript based on the reviewers' suggestions, making modifications to ensure accuracy and avoid misinterpretation.
- And if you find "identical", the specific species should be stated.
A: In Section 3.6, “identical” means that the ASVs in V3-V4 region of 16S rRNA are 100% identical as shown in Fig. 4, indicating that the clones are identical. We slightly modified the description.
- The discussion should state the bacterial names. The text is too superficial and not conclusive. The conclusion should not be an estimate, but a conclusion from the data obtained from this study.
A: If all the bacteria names were included in the first paragraph of the discussion, it will be extremely long and hard to read. In the third paragraph of the discussion, we have described each category of bacteria according to Socransky's classification, which we believe is sufficient to understand. In the fourth paragraph, bacterial names are described by genus. In the fifth paragraph, we are describing the contents of the cited article and writing the names of the bacteria described in that article. In the sixth paragraph, we describe oral management, and there is no place to include the name of the bacteria.
The reviewer's criticism is that the conclusions are superficial or estimation, but we believe that the conclusions are logically derived from the facts obtained in this study. This study is a clinical observational study, and it is not possible to show a causal relationship. It has been proven that the oral bacteria and the bacteria of the aortic valve are the same clone, which can be considered direct evidence of the association of the pathogenesis of aortic valve disease with oral bacteria. Since it is not possible to determine from this study whether the oral bacteria present in the aortic valve are involved in the onset or progression of aortic valve disease, we slightly modified the conclusion.
Round 2
Reviewer 2 Report
Comments and Suggestions for Authors
This revised version has significantly improved from the previous submission. The introduction is more substantial, and the conclusions are appropriately moderated. The study is interesting and has a potential impact. However, a few remaining minor points should still be addressed:
- Please further simplify and clarify complex figures (ASV match lists) with clearer legends or explanatory notes. Figure 3 is too extensive. I suggest compiling a figure that includes the most relevant information and moving the rest to supplementary material.
- Please expand your discussion of the sample-size limitation and the lack of healthy controls in more detail.
Author Response
1. Please further simplify and clarify complex figures (ASV match lists) with clearer legends or explanatory notes. Figure 3 is too extensive. I suggest compiling a figure that includes the most relevant information and moving the rest to supplementary material.
A: Following the reviewer's instructions, the original Fig. 3 has been converted to supplemental data (Fig. S6). New Fig. 3 now only includes data on the aortic valve, and only includes oral commensal bacteria and Sokransky’s color-coded bacteria. The text and legend descriptions have been modified.
3.5. Characteristics of bacteria detected in the tongue, dental plaque, and resected aortic valve
The 195 bacteria detected in the tongue, dental plaque, and resected aortic valve specimens were ranked according to read count (Fig S1, S2, S3, S6). The lists of the top 25 bacteria are shown in Fig S6A, S6B, and S6C. Among the bacteria detected in the resected aortic valve, oral commensal bacteria and periodontal pathogenic bacteria (color-coded as red, orange, yellow, green, blue, and purple) according to Socransky's classification [23] were extracted and shown in Fig. 3. Among the bacteria listed for resected aortic valves, 48% (16-64%) of the 35 species were associated with the oral cavity, and 18% (4-28%) were periodontal pathogenic bacteria (Fig S6A, S6B, and S6C). In cases 20 and 25, red and orange bacteria were detected in the aortic valve. In Case 30, neither red or orange complex bacteria were detected in the aortic valve (Fig. 3). In cases 14, 18, and 28, red and orange bacteria were detected in the aortic valve (Fig. 3), although the bacterial microbiota detected on the dorsal surface of the tongue contained no red and only a few orange bacteria (Fig S6C). In cases 20, 25, and 30, a significant number of red and orange bacteria were included in the bacterial microbiota considered endemic to the oral cavity detected on the dorsal surface of the tongue (Fig S6B, and S6C).
Title for Fig 3
Oral commensal bacteria and periodontal pathogenic bacteria detected in the resected aortic valve
Legend for Fig 3
Among the bacteria detected in the resected aortic valve, oral commensal bacteria and periodontal pathogenic bacteria (color-coded as red, orange, yellow, green, blue, and purple) according to Socransky's classification were extracted from the lists of the top 25 bacteria (Fig S6A) and shown in Fig 3. In cases 20 and 25, red and orange bacteria were detected in the aortic valve. In Case 30, neither red or orange complex bacteria were detected in the aortic valve. In cases 14, 18, and 28, red and orange bacteria were detected in the aortic valve.
Title for FigS6
Fig S6. Top 25 bacteria listed by read count in each sample.
Legend for Fig S6
Bacteria related to oral cavity are written in bold. The color of the background of the bacteria name was the same color as the bacteria matching Sokransky's color code in Figure 2. (A) List of aortic valve samples. (B) List of dental plaque samples. (C) List of tongue samples.
2. Please expand your discussion of the sample-size limitation and the lack of healthy controls in more detail.
A: The issue of the number of samples analyzed in this study, as well as the difficulty of analyzing the healthy controls, has already been described in Potential Limitations, paragraph 2 of the Discussion section. I have expanded on these issues.
A potential limitation of this study is whether the fact that bacterial DNA was detected in the aortic valve in only 12 of 32 cases, and that sufficient bacterial DNA was recovered for metagenomic analysis in only 6 cases, is a result of manipulation or a genuine finding. In the future, it will be necessary to quantify the percentage of patients who have the same clone of oral bacteria in the aortic valve, and to strictly control the detection limit of bacterial DNA from the valve to prevent cross-contamination. As this study is a PCR-based experiment, increasing the sensitivity reduces the specificity. Increasing the number of patients and setting a strict detection sensitivity threshold makes it possible to determine whether the detection frequency obtained in this experiment reflects the true situation or is merely a reflection of the detection sensitivity. Furthermore, since we are unable to examine the presence of oral bacteria in the aortic valves of healthy subjects or in patients with mild aortic valve disease, it is not known whether this observation is a phenomenon specific to patients with severe aortic valve disease. A study investigating the presence of bacteria in heart samples taken during autopsies of individuals without aortic valve disease who died in accidents could confirm whether the findings of this study are specific to patients with aortic valve disease.
Reviewer 3 Report
Comments and Suggestions for Authors
The authors have made substantial efforts to revise the manuscript in response to the major and minor concerns raised during the initial review. Overall, the revised version demonstrates significant improvement in methodological clarity and discussion of limitations. The manuscript now presents a compelling observational study investigating the presence of oral bacteria in aortic valves, suggesting a potential link between oral dysbiosis and aortic valve disease. While further mechanistic or longitudinal data will be required in future studies, this work provides valuable evidence for the presence of oral bacterial DNA in diseased aortic valves and reinforces the need for good oral hygiene in systemic disease prevention.
Author Response
We are very grateful for your valuable comments during the review process. This has helped us clarify the study's message.
Reviewer 4 Report
Comments and Suggestions for Authors
Comments on revised microorganisms-3713409
The authors have done most of the corrections, but some important issues are still missing.
Title: Do you have any evidence for bacteremia? This was not shown in the present paper. Actually, in line 397, it says that bacteremia is unlikely to be the cause of bacterial adherence to aortic valve. Thus, the "resulting bacteremia" can be omitted from the Title.
Abstract:
The two first sentences are overlapping, and can be made into one sentence. Line 28: It is not right to write "only a few reports". This can be deleted. Actually, it stays in contrast to Line 55, where it is stated: "several reports".
"Distal organs" can be replaced with more specific sites.
Line 29: correct to "analyzed"
Sentence in Line 36 is a repetition of sentence in Line 37 and can therefore be deleted.
Line 41: You cannot say "identical" (which means 100% similarity). Better to say "similar". Also add some of the major bacterial species observed at both sites. In the result section it says: " A significant difference in microbial composition was observed among the three groups". Thus, the abstract should reflect the data in the result section.
The last sentence in the abstract "the resulting bacteremia" was not shown in this paper, so it has to be tuned down as a possible connection.
Introduction:
The Introduction can be more comprehensive including specific information from previous studies. Some of the issues were pinpointed in the response letter, but the authors ought to have included these issues in the introduction. E.g., the oral bacteria known to cause diseases of other organs, the mechanism of how AV increases the risk of IE. Etc.
Line 72: "identical"?? or similar?
Sentence in Line 76 should be in past tense.
Methods:
Section 2.6: The software used should be stated.
Results:
In Table 1: Please correct " carification" to "calcification". If you have 100% ASV match, do you intend to the same kinds of bacteria? However, from the graphs it seems that there are different distribution and prevalence.
What do you intend by: " and 32 patients were not extracted from 2378 patients".
Figure 1: The Y-axis lacks a title label. And the figure lacks a definition of the two grey colors of the bars. The letters in the graph should be in black. (Currently it is in weak grey color).
Line 291: State that the antibody titer of healthy individuals was set to 100.
Figure 2: You do not need the two digits after the point of the numbers in Y-axis. The letters should be in black. X-axis should have a title such as "Stage of Periodontitis".
Figure 4: Although obvious, P, T and V should be defined in the legend.
Supplementary Figure S1: You need to enlarge the letter size of the bacterial names. Also, there should be a synchronization of the columns and the labels of each column. The labels are not exactly under each own column except for the first ones.

Author Response
1. The authors have done most of the corrections, but some important issues are still missing.
A: I really appreciate your detailed reading and critical comments. I will revise the paper according to your suggestions to further improve the quality of the paper.
2. Title: Do you have any evidence for bacteremia? This was not shown in the present paper. Actually, in line 397, it says that bacteremia is unlikely to be the cause of bacterial adherence to aortic valve. Thus, the "resulting bacteremia" can be omitted from the Title.
A: As pointed out, we have no direct evidence that bacteremia is occurring; unlike IE patients, we do not routinely perform blood bacterial cultures when treating aortic valve patients. However, there is no other route for oral bacteria to reach the aortic valve except through the blood circulation. As the reviewer pointed out, we have removed the word “bacteremia” from the title and added “the pathogenesis of”. In discussion and conclusion, I would like to add the word “potential” to the sentence and leave the word bacteremia.
Title
Oral dysbiosis is associated with the pathogenesis of aortic valve diseases
For abstract, discussion, and conclusion
Oral dysbiosis and the resulting potential bacteremia are associated with the pathogenesis of aortic valve diseases.
Line 397: oral bacteria invade the body and settle in the aortic valve via the bloodstream.
This statement means that oral bacteria can potentially cause bacteremia and reach the aortic valve. I have modified it slightly to make it easier to understand.
Oral bacteria continuously invade the bloodstream and settle in the aortic valve.
Abstract:
3. The two first sentences are overlapping, and can be made into one sentence. Line 28: It is not right to write "only a few reports". This can be deleted. Actually, it stays in contrast to Line 55, where it is stated: "several reports".
A: As you mentioned, it is overlapping. Similar studies like ours were not evident when we closed this study. We found several papers when we started submitting our manuscript and modified the text, although this part was left uncorrected. We scratch off the second sentence from the abstract.
4. "Distal organs" can be replaced with more specific sites.
A: We added the actual organs in the text.
The involvement of oral bacteria in the pathogenesis of distant organs, such as heart, lungs, brain, liver, and intestine, has been shown.
5. Line 29: correct to "analyzed"
A: We corrected.
6. Sentence in Line 36 is a repetition of sentence in Line 37 and can therefore be deleted.
A: We deleted the redundant part “Most patients with aortic valve disease have severe periodontal disease.”.
7. Line 41: You cannot say "identical" (which means 100% similarity). Better to say "similar". Also add some of the major bacterial species observed at both sites. In the result section it says: " A significant difference in microbial composition was observed among the three groups". Thus, the abstract should reflect the data in the result section.
A: As the reviewer pointed out, whole genome sequencing is necessary to prove the identity of the bacterial clones. Therefore, we believe that the phrase “The genomic sequences of the V3-V4 region of the 16S rRNA in some bacteria isolated from the aortic valves of 6 patients who underwent metagenomic analysis were identical to those found in the oral cavity.” is appropriate. The expression of bacteria will be changed to “strain-level similarity” throughout the paper, as we interpret this result. We also change “3.6. Identification of the same bacterial clones in the tongue, dental plaque, and resected aortic valve” to “Identification of the same ASV in the V3 -V4 region of the 16S rRNA sequence in the tongue, dental plaque, and resected aortic valve”. We consider this finding to be the most noteworthy result of this paper, and therefore we have chosen to express it cautiously.
The genomic sequences of the V3-V4 region of the 16S rRNA in some bacteria isolated from the aortic valves of 6 patients who underwent metagenomic analysis were identical to those found in the oral cavity.
3.6. Identification of the same ASV in the V3 -V4 region of the 16S rRNA sequence in the tongue, dental plaque, and resected aortic valve strain-level similarity
8. The last sentence in the abstract "the resulting bacteremia" was not shown in this paper, so it has to be tuned down as a possible connection.
A: We rearranged our expression
Oral dysbiosis and the resulting potential bacteremia are associated with the pathogenesis of aortic valve diseases.
Introduction:
9. The Introduction can be more comprehensive including specific information from previous studies. Some of the issues were pinpointed in the response letter, but the authors ought to have included these issues in the introduction. E.g., the oral bacteria known to cause diseases of other organs, the mechanism of how AV increases the risk of IE. Etc.
A: In response to the reviewer's suggestion, we have added the mechanism by which AV diseases increase the risk of IE in the Introduction. We have cited the previously published article and mentioned that oral bacteria can cause diseases in distant organs. I specifically mentioned “distant organ” as pointed out by other reviewers.
In general concept of the pathogenesis of IE, the blood flow jets caused by the valvular diseases (AS and AR) damage the valves and endocardium, resulting in the formation of a sterile vegetation at the site of the damage. Bacterial infection on these vegetations is thought to lead to the development of IE.
10. Line 72: "identical"?? or similar?
A: The sequence of the entire 16S rRNA region is 100% identical. Therefore, it is “identical”. We added 100% in the text.
11. Sentence in Line 76 should be in past tense.
A: We corrected.
Methods:
12. Section 2.6: The software used should be stated.
A: We added the name of the software as follows. “Before analysis by Qiime2,”
Results:
13. In Table 1: Please correct " carification" to "calcification". If you have 100% ASV match, do you intend to the same kinds of bacteria? However, from the graphs it seems that there are different distribution and prevalence.
A: We corrected misspelling. Table 4 presents the read counts of bacteria with matching ASVs at each site (aortic valve, dorsal surface of the tongue, and dental plaque). The read counts of bacteria with matching AVS at each site (aortic valve, dorsal surface of the tongue, or periodontal pocket) are shown. When an ASV of V 3-V4 16S rRNA at aortic valve matched those at one or two sites (aortic valve or oral cavity), it was assessed as “strain-level similarity”. Of course, which bacteria were detected and which ASVs were matched varied from case to case.
14. What do you intend by: " and 32 patients were not extracted from 2378 patients".
A: The 32 are not drawn from the population of 2,378, but from a different group, meaning that they are a separate group.
15. Figure 1: The Y-axis lacks a title label. And the figure lacks a definition of the two grey colors of the bars. The letters in the graph should be in black. (Currently it is in weak grey color).
A: Thank you for pointing this out. It was a careless mistake. I changed the text to black. The Y-axis shows the incidence of periodontal disease severity, and I used percentages for clarity. I also left out what each column indicates. A legend has been written in the graph.
16. Line 291: State that the antibody titer of healthy individuals was set to 100.
A: Thank you for pointing this out, Line 281, I have added the following.
The antibody titer of healthy individuals was set to 100.
17. Figure 2: You do not need the two digits after the point of the numbers in Y-axis. The letters should be in black. X-axis should have a title such as "Stage of Periodontitis".
A: We have corrected it as suggested.
18. Figure 4: Although obvious, P, T and V should be defined in the legend.
A: Yes, we added it in the legend.
P: dental plaque, T: dorsal surface of tongue V: aortic valve.
19. Supplementary Figure S1: You need to enlarge the letter size of the bacterial names. Also, there should be a synchronization of the columns and the labels of each column. The labels are not exactly under each own column except for the first ones.
A: We have corrected it as suggested.